# Soil Mercury Pollution Changes Soil Arbuscular Mycorrhizal Fungal Community Composition

**DOI:** 10.3390/jof9040395

**Published:** 2023-03-23

**Authors:** Yidong Mi, Xue Bai, Xinru Li, Min Zhou, Xuesong Liu, Fanfan Wang, Hailei Su, Haiyan Chen, Yuan Wei

**Affiliations:** 1College of Environment, Hohai University, Nanjing 210098, China; 2State Key Laboratory of Environmental Criteria and Risk Assessment, Chinese Research Academy of Environmental Science, Beijing 100012, China; 3Department of Administration Service, Ministry of Ecology and Environment of the People’s Republic of China, Beijing 100006, China

**Keywords:** arbuscular mycorrhizal fungi (AMF), mercury, heavy metal, community composition, bioremediation

## Abstract

Remediation of mercury (Hg)-contaminated soil by mycorrhizal technology has drawn increasing attention because of its environmental friendliness. However, the lack of systematic investigations on arbuscular mycorrhizal fungi (AMF) community composition in Hg-polluted soil is an obstacle for AMF biotechnological applications. In this study, the AMF communities within rhizosphere soils from seven sites from three typical Hg mining areas were sequenced using an Illumina MiSeq platform. A total of 297 AMF operational taxonomic units (OTUs) were detected in the Hg mining area, of which Glomeraceae was the dominant family (66.96%, 175 OTUs). AMF diversity was significantly associated with soil total Hg content and water content in the Hg mining area. Soil total Hg showed a negative correlation with AMF richness and diversity. In addition, the soil properties including total nitrogen, available nitrogen, total potassium, total phosphorus, available phosphorus, and pH also affected AMF diversity. Paraglomeraceae was found to be negatively correlated to Hg stress. The wide distribution of Glomeraceae in Hg-contaminated soil makes it a potential candidate for mycorrhizal remediation.

## 1. Introduction

Mercury (Hg) has been considered a global pollutant due to its high toxicity, persistence, potential for bioaccumulation, and long-distance transmission [1,2,3,4]. Soil pollution by Hg is a serious problem resulting from both natural [5,6] and anthropogenic activities [7,8]. Fossil fuel combustion [9,10], metal mining, refining, and manufacturing [11,12] are the major soil Hg pollution sources by human activities. In addition, soil Hg can accumulate in food chains, potentially threatening human health [13,14]. In European Union countries, 200,000–1,800,000 newborns suffer from intellectual impairment each year due to dietary exposure to Hg [14]. In China, there was a decrease of 0.14 points in the IQ of newborns and 7360 deaths from fatal heart attacks linked to Hg intake in 2010 [15]. The annual global release of Hg from human activities was estimated as 2000 Mg, whereas the anthropogenic emissions of Hg in China reached 538 Mg in 2010 [16]. At present, East Asia, especially China, has become the most serious soil Hg-contaminated area in the world [17]. The development of ecological remediation is urgent and has attracted much attention in Hg-contaminated areas.

Phytoremediation is an important method in ecological remediation, and shows great potential in Hg-contaminated soil because of its high efficiency and low cost [16,18,19]. The use of Hg-enriching plants to extract Hg from Hg-contaminated soils (phytoextraction) [16,20] or the growth of Hg-tolerant plants in Hg-contaminated soils (phytostabilization) [21] are potential soil phytoremediation strategies. Some native soil microorganisms, such as arbuscular mycorrhizal fungi (AMF), can improve plant adaptability and growth status in this process, further improving the efficiency of the phytoremediation of heavy metal soils [22,23,24]. However, the lack of studies on the correlation between AMF community composition and soil Hg contamination has hindered the effective application of mycorrhizal technology in Hg-contaminated areas.

AMF are ubiquitous in soil environments and interact with most plants through mycorrhizal structures [25]. AMF are important plant performance regulators in heavy metal-contaminated soils and can increase plant resistance to heavy metal toxicity through mycorrhizal filtering [26], stabilization [27], and other direct mechanisms [28,29]. AMF can also enhance the host plant’s antioxidant capacity and nutrient absorption [30], as well as regulate rhizosphere environmental characteristics [31], and the expression of compound transporter genes [32]. The AMF-enhanced phytoremediation of heavy metal-contaminated soil has received significant attention because of its environmentally friendly characteristics. This method has been demonstrated to be effective in various heavy metal-contaminated soil remediation efforts [28,33,34]. The effects of AMF on Hg within soil–plant systems were studied by Yu et al. [35] for the first time in 2010, providing evidence for the role of mycorrhizae in increasing the immobilization of Hg in soils and reducing Hg uptake by roots. Furthermore, other studies have shown that AMF inoculation can enhance Hg immobilization in soil [36], alter Hg uptake in plants [37,38], and enhance the root elongation and seedling development of host plants [39]. In addition to AMF affecting heavy metal absorption in plants, the concentrations of heavy metals affect the distributions and community composition of AMF, such as Mn [29], Ni [40], Cu [40], Zn [41,42,43], As [44], Cd [40], Sb [25,45], and Pb [41,42,46]. Nevertheless, comparative analyses of AMF community composition in Hg-contaminated soils are lacking. This knowledge gap has become an obstacle for the biotechnological application of AMF to Hg-contaminated soil. Systematically investigating AMF community diversity and its influencing factors in Hg-contaminated soils is essential for identifying Hg-tolerant AMF and developing efficient phytoremediation techniques.

The Wanshan Hg mining area located in Guizhou Province is the world’s third largest Hg mine. The mine ceased production and closed in 2001. However, historical mining and smelting activities in the Wanshan Hg mining area have caused serious Hg pollution that threatens the health of residents near Hg mining areas [47,48]. This makes the Wanshan Hg mining area an ideal system for evaluating AMF community compositions in Hg-contaminated soil systems. Four plants (*Eleusine indica*, *Artemisia argyi*, *Astragalus sinicus*, and *Conyza canadensis*) that are widely distributed in Wanshan Hg mining areas were selected as the target plants for this study. In addition, *E. indica* has been confirmed to be widely distributed in Hg-contaminated areas [49]. *A. argyi* and *C. canadensis* have higher tolerance of Hg, with a bioaccumulation factor for Hg up to 30% of *A. argyi* [50]. AMF can alleviate the As stress and regulate the growth condition of *A. sinicus*, but the resistance of *A. sinicus* to Hg is unknown [51].

In this study, AMF community composition was characterized using high-throughput sequencing technology in three typical Hg mining areas with different Hg concentrations. This study hypothesized that (1) soil Hg contamination altered soil AMF community composition and negatively affected AMF diversity; and (2) AMF from different families have different sensitivity to Hg. To address these hypotheses, this study (1) determined the dominant families, richness, and diversity of AMF in rhizosphere soils of dominant plant species (*E. indica*, *A. argyi*, *A. sinicus*, and *C. canadensis*) in different sampling sites; (2) compared the differences in AMF richness, diversity, and community composition among different host plants and different sampling sites; (3) identified the principal components (PCs) of soil properties that significantly affected AMF community composition, and quantified the corresponding responses of AMF richness and diversity to elevated soil Hg content; and (4) recognized the correlations between the AMF of different taxa and different soil properties. The results are expected to provide theoretical support for phytoremediation in Hg-contaminated areas.

## 2. Materials and Methods

### 2.1. Study Area and Soil Sampling

The samples were collected in the Wanshan Hg mining area, which is situated in the Wanshan district of Tongren City within the southern Hunan–Guizhou Hg mine belt at the eastern edge of Guizhou Province. The research area exhibits highly variable temperatures and has a subtropical humid monsoon climate, with a mean annual temperature of 13.7 °C and mean summer rainfall of 1379 mm.

Samples were collected in April 2019. Three typical Hg mining areas were selected as the research area, and each Hg mining area was divided into 2–3 sampling sites according to their mining function, as described in Figure 1 and Appendix A. Four dominant plants widely distributed in the mining area were selected for collection, namely *A. argyi*, *E. indica*, *A. sinicus*, and *C. canadensis*; the plant species collected at each sampling point are shown in Appendix A. A 20 × 20 m^2^ plot with evenly distributed plant species and soil properties was selected at each sampling site. Three individuals from each plant that exhibited vigorous growth were selected for the collection of 1 kg rhizosphere soil, which was mixed into a combined sample, and then divided into two subsamples. One subsample was refrigerated in liquid nitrogen and sent immediately to the laboratory for molecular analysis, while the other was used to analyze soil properties after air-drying. Arbuscular mycorrhizal colonization was detected in all the studied plant species.

### 2.2. Measurement of Soil Properties

Total Hg (THg) was measured using an Hg vapor meter (HGA-100, Beijing Haiguang Instrument Co., Ltd., Beijing, China). Total nitrogen (TN) was determined using the Kjeldahl method. Briefly, 1 g of air-dried soil was moistened with 1 mL of water, 2 g of catalyst (K_2_SO_4_:CuSO_4_·5H_2_O = 10:1) was added, and the sample was digested with 5 mL of H_2_SO_4_. The digestion product was distilled with 20 mL 10 mol/L NaOH solution, and the distillation product was absorbed with 10 mL of H_3_BO_3_ indicator, and titrated with 0.02 mol/L of HCl solution. Available nitrogen (AN) was measured as follows: 1 g of air-dried soil was first alkalized with 10 mL of 1.8 mol/L NaOH solution, and then the alkalized products were absorbed with 3 mL of H_3_BO_3_ indicator, and titrated with 0.02 mol/L of HCl solution. Total phosphorus (TP) was estimated by melting the soil with NaOH at 720 °C and using the molybdenum-antimony colorimetric method. Extraction with NH_4_F-HCl solution (pH < 6.5) or NaHCO_3_ solution (pH > 6.5) and estimation using the molybdenum-antimony colorimetric method was used to determine available phosphorus (AP). Total potassium (TK) and available potassium (AK) were measured with an inductively coupled plasma optical emission spectrometer (ICP-OES, 5110, Agilent Technologies, Santa Clara, CA, USA) after dissolution and ammonium acetate extraction. Soil pH was determined by adding 25 mL of water to 10 g of air-dried soil, and then shaking, standing, and estimating with a pH meter. Soil total organic carbon (TOC) was measured using the potassium dichromate oxidation spectrophotometric method. Soil water content (WC) was measured by drying soils at 105 °C to a constant weight followed by weighing.

### 2.3. AMF Community Analysis

Total DNA was extracted from 0.25 g soil with the E.Z.*n*.A.^®^ soil DNA Kit (Omega Bio-tek, Norcross, GA, USA) according to the manufacturer’s instruction. The 18S rRNA of AMF was subjected to a two-step nested PCR. The PCR program and reaction system components are shown in Appendix A. AML1: 5′-ATCAACTTTCGATGGTAGGATAGA-3′ and AML2: 5′-GAACCCAAACACTTTGGTTTCC-3 were used as primers in the first-round PCR, and AMV4.5NF: 5′-AAGCTCGTAGTTGAATTTCG-3′ and AMDGR: 5′-CCCAACTATCCCTATTAATCAT-3′ were used as primers in the second-round PCR.

PCR products were purified using the AxyPrep DNA Gel Extraction Kit (Axygen Biosciences, Union City, CA, USA) and quantified using QuantiFluor™-ST (Promega, Madison, WI, USA). Purified amplicons were sequenced using an Illumina MiSeq platform (Majorbio, Shanghai, China) via 2 × 250 bp paired-end sequencing. Raw reads have been submitted to the NCBI Sequence Read Archive (SRA) database with the following accession number: PRJNA735324.

Sequences were demultiplexed as fastq files and quality-filtered using the QIIME pipeline (version 1.17) [52]. Sequences were clustered into operational taxonomic units (OTUs) at the 97% sequence similarity, and the chimeric sequences were removed using the UPARSE program (version 7.1, http://drive5.com/uparse/, accessed on 11 June 2019). The taxonomic classification of each 18S rRNA gene sequence was analyzed with the Ribosomal Database Project classifier (http://rdp.cme.msu.edu/, accessed on 11 June 2019) against the MaarjAM 18S rRNA database [53].

### 2.4. Statistical and Data Analysis

Data analyses were performed with the SPSS 17.0 software package. Observed richness (Sobs), Shannon, Shannoneven, and Coverage indexes were calculated using Mothur (Version 1.80.2; https://mothur.org/wiki/calculators/, accessed on 11 June 2019). Sobs is the observed values of richness. The Shannon index was calculated according to Shannon and Weaver [54]. The Shannoneven index was calculated according to Pielou [55]. The Coverage index was calculated according to Good [56]. The rarefaction curve was calculated using Mothur to calculate the AMF community indexes under different samplings using the R package. The AMF community index bar charts were constructed using Origin (8.0). The community bar charts were constructed using the R package based on sequencing data tables. The significance levels (F values) of the effects of plants, sampling sites, and their interactions on the AMF community indexes were tested using one-way analysis of variance (ANOVA) analysis, and the least significant difference (LSD) test was used to compare averages at the 0.05 probability level (*p* = 0.05). Differences among the means of soil properties and AMF community indexes were tested using *t* tests. Principal coordinates analysis (PCoA) was employed to illustrate different soil sample clustering using the R package based on the Bray–Curtis distance. Permutational multivariate ANOVA (PERMANOVA) based on the Bray–Curtis distance were conducted to analyze the significant difference in the AMF community compositions among different sampling sites or different plants. In order to avoid multicollinearity, principal component analysis (PCA) was performed using the SPSS 17.0 software package. The derived orthogonal principal components (PCs) were then used to explain variance in the AMF communities. The correlation between PCs and AMF communities was elucidated through canonical correlation analysis (CCA) using the R package. General linear mixed models were using to model the AMF community indexes against the soil PCs in the SPSS 17.0 software package. Linear regression was performed using Origin (8.0). The Mantel test was used to compare the correlations between soil properties and the Sobs, Shannon, and Shannoneven indexes of the AMF community composition in R using the “ggcor” package. The heatmaps were created using the R package and the correlations were obtained based on the Pearson correlation coefficient. Two-factor correlation network analysis was performed using Gephi 0.9.

## 3. Results

### 3.1. Overall Sequencing Results and AMF Community Composition

After leveling out according to the minimum number of sample sequences, a total of 1,263,408 sequences matched with Glomeromycota after quality filtering. Among them, 297 OTUs were detected in the Wanshan Hg mining area, and a greater number of OTUs belonged to the Glomeraceae (66.96%, 175 OTUs, 845,921 sequences). Others belonged to the Diversisporaceae (28.63%, 24 OTUs, 361,661 sequences), Acaulosporaceae (2.05%, seven OTUs, 25,959 sequences), and Paraglomeraceae (1.76%, 13 OTUs, 22,289 sequences), while only a small number belonged to Archaeosporaceae (0.25%, five OTUs, 3130 sequences) and Gigasporaceae (<0.01%, one OTU, one sequences). In addition, 0.35% of all sequences were unclassified at the family level. Detailed information on the OTUs in each sampling site is provided in Appendix A.

Almost all rarefaction curves tended to saturate at the chosen sequencing depth (Appendix A). The AMF Sobs, Shannon, Shannoneven, and Coverage indexes were shown in Appendix A. The Sobs index of each sample in the Wanshan Hg mining area ranged from 52 to 120 OTUs, and the Shannon and Shannoneven indexes ranged from 1.42 to 3.46 and from 0.33 to 0.76, respectively. The Coverage index of all samples was higher than 0.99. According to the result of one-way ANOVA (Table 1), there were significant differences in the Sobs and Shannon indexes among different sampling sites (F = 6.742, *p* = 0.001 on the Sobs index; F = 3.562, *p* = 0.018 on the Shannon index), but no differences among different plant species (F = 0.273, *p* = 0.844 on the Sobs index; F = 0.306, *p* = 0.821 on the Shannon index).

The AMF community composition at the family level across all sampling sites was explored (Figure 2A). The β-diversity of the AMF community was illustrated with PCoA plots based on the Bray–Curtis method, with different sampling sites (Figure 2B) and different plants (Figure 2C) as the basis for classification. There were significant differences between different sampling sites, but no significant differences between different plants, which was confirmed by PERMANOVA (Appendix A).

### 3.2. Correlation Analysis of Environmental Factors and AMF Community Composition

A large range of THg was detected in the soil at sampling sites in the Hg mining area (Table 2). More specifically, the THg across soil samples from the Hg mining area ranged from 1.90 to 105.00 mg/kg. As many soil variables were correlated, the soil property data of the Hg mining area were reduced in their dimensions through PCA. Three PCs were extracted that together explained 70.54% of the variance in the soil variables (Table 2). PC1 explained 38.53% of the total variance and was positively correlated with AP, AN, TK, TP, TN, and TOC, and negatively correlated with pH. PC2 explained 17.29% of the variance with positive correlations with AP and TP. PC3 explained 14.72% of the variance and was positively correlated with THg and negatively correlated with WC. CCA with forward selection showed a relationship between all PCs and the AMF community composition (Figure 3, Appendix A). All PCs explained 15.3% of the variance in the AMF communities. PC1 (R^2^ = 0.617, *p* = 0.001) and PC3 (R^2^ = 0.609, *p* = 0.001) significantly affected the AMF community composition (Appendix A).

The general linear mixed models revealed that PC3 was significantly negative correlated with the Sobs index (*p* = 0.009), the AMF Shannon index (*p* = 0.001), and the Shannoneven index (*p* = 0.001), and PC2 was negatively correlated to the Shannoneven index (*p* = 0.041) (Appendix A). To further clarify the relationship between PCs and the AMF richness and diversity in Hg-contaminated sites, linear regression analysis of three PCs and the Sobs, Shannon, and Shannoneven indexes of AMF was conducted (Figure 4). A negative relationship was found between the PC3 and the Sobs index (R^2^ = 0.284, *p* = 0.007 < 0.01), between PC3 and the Shannon index (R^2^ = 0.417, *p* = 0.001), and between PC3 and the Shannoneven index (R^2^ = 0.360, *p* = 0.002 < 0.01). A Mantel test was applied to calculate the correlation between soil properties and the Sobs, Shannon, and Shannoneven indexes (Figure 5). TAs was significantly negatively correlated with WC, while TP had a significant positive correlation with AP. There were significant correlations among AN, TOC, TN, and TK, and they were negatively correlated with pH. The Sobs index was correlated with THg (R = 0.227; *p* = 0.027) and AK (R = 0.314, *p* = 0.002); Shannon (R = 0.490, *p* = 0.012) and Shannoneven (R = 0.393, *p* = 0.030) indexes were correlated with TP. Further analysis of Pearson correlation coefficients between soil properties and AMF richness and diversity indexes revealed that THg had a significant negative correlation with the Sobs index and negatively but not significantly correlated with Shannon and Shannoneven indexes (Appendix A).

### 3.3. Correlation Analysis of Environmental Factors and AMF Community Composition

A correlation heatmap was used to demonstrate the correlation between soil properties and AMF abundance on the family level in the Hg mining area (Appendix A). The abundance of Paraglomeraceae was significantly negatively correlated with the THg. Two-factor network analysis of the associations between the AMF OTUs of different families and soil properties was performed in the Hg mining area (Figure 6), and the results showed that a total of 18 OTUs were correlated with THg, indicating that THg was particularly important for the distribution of the AMF community composition, followed by TN (16 OTUs), TOC (12 OTUs), AN (11 OTUs), WC (nine OTUs), AK (eight OTUs), pH (eight OTUs), TP (seven OTUs), and TK (seven OTUs).

## 4. Discussion

### 4.1. Dominant AMF Families in Hg-Contaminated Soils

To date, the AMF communities of Hg-contaminated soils have not been studied, and the sensitivity and tolerance of AMF community composition to soil Hg content remain unclear, which makes it difficult to effectively conduct research on the application of AMF in the ecological remediation of Hg-contaminated soil and the migration and transformation of Hg in soil–plant systems. This study characterized AMF communities using a high-throughput sequencing approach in the Wanshan Hg mining area to confirm the hypothesis that the AMF community composition was affected by soil Hg pollution. Although the Hg mining area in Wanshan has already been closed for decades, THg levels still far exceed the threshold value for identifying soil Hg pollution risk within the construction industry soils of China (38 mg/kg) [57] (Table 2), which is in agreement with previous studies [58,59].

A total of 1,263,408 sequences and 297 OTUs were obtained in the Hg mining area based on the MaarjAM database. Among them, 66.96% of sequences (175 OTUs) were identified as Glomeraceae, which indicated that Glomeraceae was the dominant family in Hg-contaminated soils. Glomeraceae is dominant in most soil ecosystems. Numerous studies have suggested that Glomeraceae exhibits strong ecological adaptability and is the dominant genus in many ecosystems, even in areas with heavy metal stress [25,41,43,46,60,61,62]. One reason is that Glomeraceae can repair damaged hyphae in stressful environments, which helps to improve their ability to adapt to environmental stresses [63,64]. Another reason is that plants harbor Glomeraceae that can benefit their growth and improve their competitiveness under specific environmental conditions [45,64], thereby promoting the dominance of Glomeraceae in soil ecosystems. Other studies suggest that the predominant detection of Glomeraceae might be explained by the higher sporulation rate [62]. Previous studies have demonstrated that Glomeraceae can form beneficial symbiotic relationships with various plants, and enhance the efficiency of ecological restoration of heavy metal-contaminated soil through promoting the growth or heavy metal absorption of plants [28]. Glomeraceae is widely distributed, even in the Hg-polluted soil in the study area, and seems to be a potential option to promote ecological remediation efficacy in Hg-contaminated areas. Diversisporaceae also had a relatively high abundance in the sampling area, with 24 OTUs (28.63% of all sequences). This AMF family is also widely distributed in various ecosystems [65,66,67,68,69].

### 4.2. Soil Hg Concentration Induced Changes in AMF Community Composition

Although Glomeraceae was the dominant family in the research area, the richness and diversity of AMF in all sampling sites differed significantly (Appendix A). Previous studies have demonstrated that the compositions of AMF varied between soils with different levels of heavy metal pollution [42,46,70], consistent with the results of this study (Figure 2B), which may be the result of the compound effects of various soil indexes. Indeed, the community composition of AMF is directly affected by soil properties [66,71,72,73,74]. Some studies have suggested that host plants could influence the rhizosphere AMF community composition [75,76], and other studies have found that compared with host plants, the AMF community composition is more strongly affected by soil properties [42,46]. The results of this study showed that spatial differences in soil properties had a more significant impact on AMF communities and diversity than differences in host plant species in the focus area (Table 1 and Appendix A, Figure 2C).

In this study, AMF compositions were found to be significantly influenced by PC1 (this PC was positively correlated with AP, AN, TK, TP, TN, and TOC, and negatively correlated with pH) and PC3 (this PC was positively correlated with THg and negatively correlated with WC) (Figure 3, Table 2) in the Hg mining area. As verified by the general linear mixed model, PC3 was negatively correlated to AMF richness and diversity. The mantel test showed that THg was negatively correlated with the Sobs index and negatively correlated with Shannon and Shannoneven indexes (Figure 5). Two-factor network analysis further confirmed that THg was the dominant factor affecting the AMF community (Figure 6). This result was in agreement with several studies that reported a negative correlation between the soil heavy metal concentration and AMF diversity [46,61]. Schneider et al. [44] and Parvin et al. [61] both observed that the abundances, species richness, and diversity of AMF decreased with increasing soil arsenic concentrations. The same trend was found in the correlation between antimony and AMF diversity [25,45]. Wei et al. [29] found that AMF diversity was significantly negatively correlated with soil Mn content. High-soil heavy metals concentrations can also inhibit spore germination and AMF mycelium growth, thereby reducing spore densities and AMF species abundances [41,61]. Excessive heavy metal concentrations are toxic to most AMF, with only AMF exhibiting strong tolerances to enable survival, thereby reducing AMF diversity [46,77].

Hg content was the dominant factor affecting the AMF community in the Hg mining area was investigated in this study, and the effects of Hg were confirmed to be negative. However, the other soil properties of the different study sites varied greatly (Table 2), which may have been another factor influencing the AMF diversity (Figure 3). Soil factors can exert profound, yet variable, influences on AMF [67,78]. Previous studies have shown that AMF diversity is affected by the soil fertility, for which the *n* content cannot be ignored. Excessive *n* input can dramatically alter the AMF community [79], which may drive AMF responses due to resource availability [80,81]. Two-factor network analysis supported this possibility (Figure 6). Sixteen OTUs and 11 OTUs were correlated with TN and AN, respectively, and *n* was closely related to the AMF community composition. Potassium has also been shown to affect AMF communities [82,83].

In addition, previous studies have shown that phosphorus is a crucial factor influencing the AMF community, and too much available phosphorus inhibits AMF community development [42]. In this study, TP and AP were correlated with PC1 and PC2 (Table 2), respectively, and PC2 was negatively correlated with Shannoneven. Further correlation analysis between single soil properties and AMF richness and diversity showed that AP was negatively correlated with Shannon and Shannoneven, while TP was negatively correlated with Shannoneven (Figure 5, Appendix A). There are reciprocal reward mechanisms between AMF and host plants [84], wherein AMF can provide mineral nutrients to host plants (especially P) and other benefits, including protection against biotic and abiotic stresses [64,85]. In exchange, plants supply AMF with carbohydrates, which are essential for fungal survival and growth [86]. High-soil P content allows plants to take up P without relying on mycorrhizal symbionts, which in turn reduces the supply of carbohydrates from the host plants to the symbiotic AMF [87]. Finally, AMF diversity decreases due to the competition for carbohydrates between AMF species [64].

### 4.3. Relationships between AMF Families and Hg

In the present study, Paraglomeraceae has been found to be negatively correlated with THg, and positively correlated with WC. Similar findings were also obtained in the study carried out by Ban et al. [88], in which Paraglomeraceae were detected at higher relative abundances in areas with low heavy metal content and high moisture content. Faggioli et al. [46] also found that Paraglomeraceae presented opposite responses to increasing soil heavy metal concentrations. However, there is no sufficient evidence to confirm whether heavy metals or soil moisture content caused the difference in the relative abundance of Paraglomeraceae. Differential relationships of Glomeraceae species to heavy metal contamination have been reported [46]. In the present study, three OTUs of Glomeraceae were negatively correlated and eight OTUs of Glomeraceae were positively correlated with THg (Figure 5). Even among AMF in the same family, the relationship with Hg varied.

## 5. Conclusions

In this study, high AMF diversity was found in soils of the Wanshan Hg mining area, and a total of 297 AMF OTUs were identified, of which the majority belonged to Glomeraceae. The diversity of AMF was negatively correlated with soil THg. One possible reason was that Hg pollution greatly disturbed the ecological composition of AMF, and as Hg pollution increased, Hg-intolerant AMF began to struggle to survive. In addition, the soil properties, including TN, AN, TK, TP, AP, and pH, could also affect AMF diversity. Though Glomeraceae was the dominant family in the Hg mining area, Paraglomeraceae was found to be more sensitive to Hg stress. These results can be considered a reference for future phytoremediation strategies based on the selection and utilization of native AMF species that are tolerant to soil Hg contamination. However, the affinity and promotion effects of AMF from different families with different plant species under Hg contamination should be further verified.

## Figures and Tables

**Figure 1 jof-09-00395-f001:**
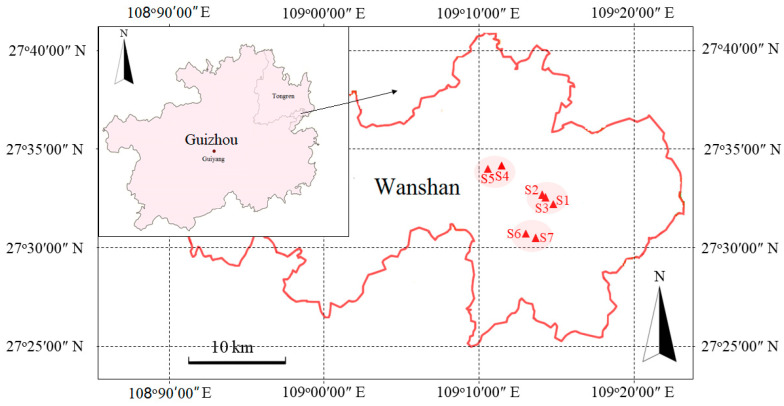
All sampling sites in the Wanshan Hg mine area of Guizhou province are indicated with triangles. The small map shows the location of the Wanshan district in Guizhou Province, China. The large map shows the layout of each sampling point.

**Figure 2 jof-09-00395-f002:**
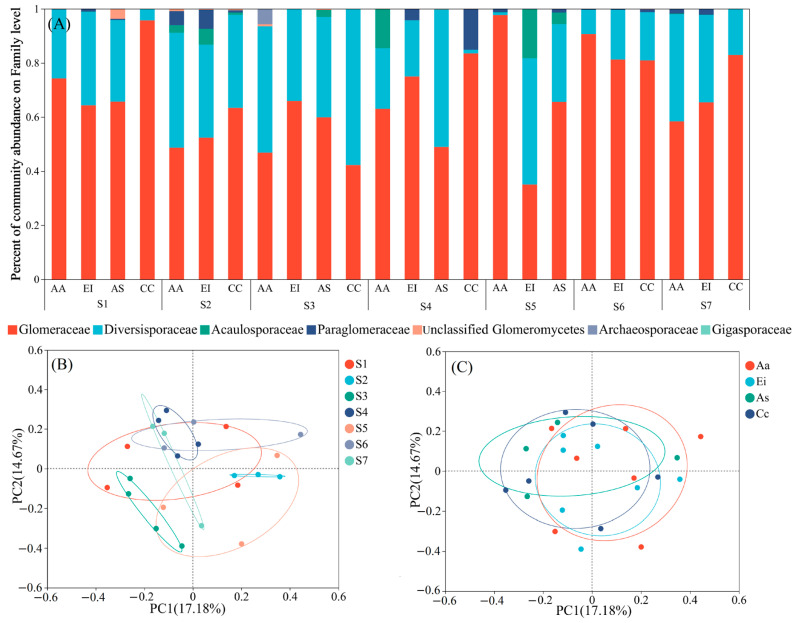
Relative abundances of reads of detected families of Glomeromycota in all sampling sites (**A**) and principal coordinates analysis (PCoA) plots of communities of arbuscular mycorrhizal fungi (AMF) across the four sampling sites based on the Bray–Curtis method with different sampling sites (**B**) (R = 0.305, *p* = 0.003) and different plants (**C**) (R = 0.037, *p* = 0.693) as the basis for classification. Aa: *Artemisia argyi*, Ei: *Eleusine indica*, As: *Astragalus sinicus*, and Cc: *Conyza canadensis*.

**Figure 3 jof-09-00395-f003:**
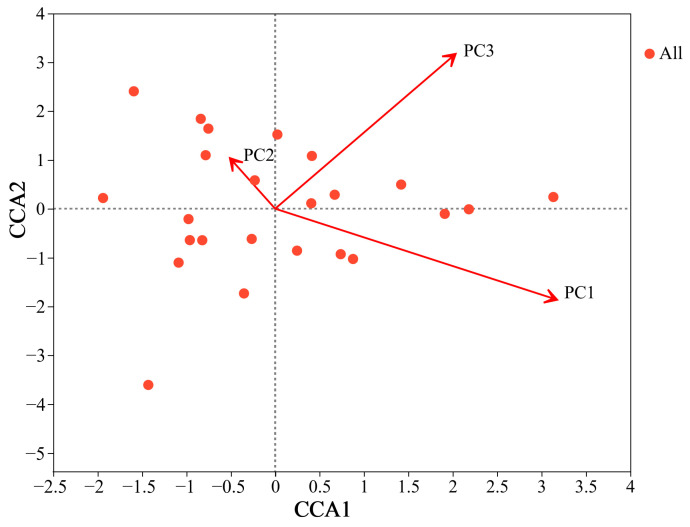
Canonical correspondence analysis (CCA) ordination plots of the relationship between all principal components (PCs) and arbuscular mycorrhizal fungi (AMF) community composition based on Bray–Curtis distance. Correlation between PCs and individual soil variables is presented in Appendix A The CCA1 and CCA2 axes accounted for the 34.3% and 31.4% of the total explained variation in AMF communities, respectively. The red dots are all samples collected in this study.

**Figure 4 jof-09-00395-f004:**
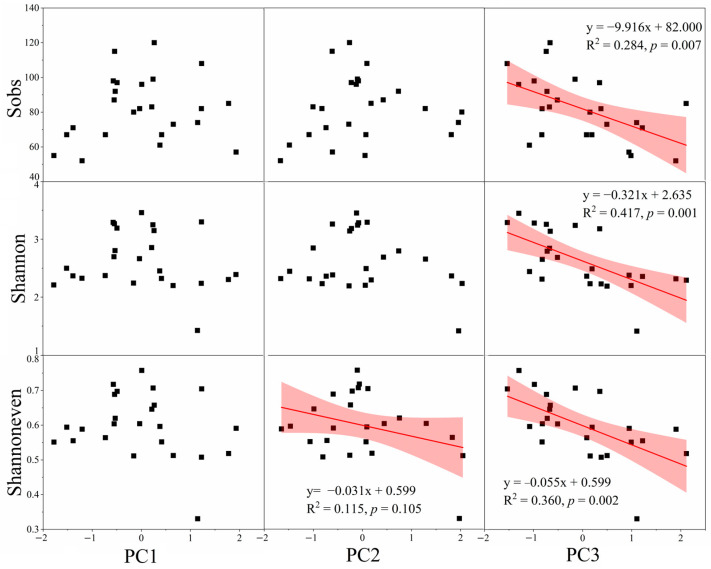
Relationship of the Sobs, Shannon, and Shannoneven indexes with soil principal components (PC) 1, PC2, and PC3 across samples from the Hg mining area. The red line is the fitting curve of the scatter plot, and the shaded area is the confidence region (95%) for the fitted line.

**Figure 5 jof-09-00395-f005:**
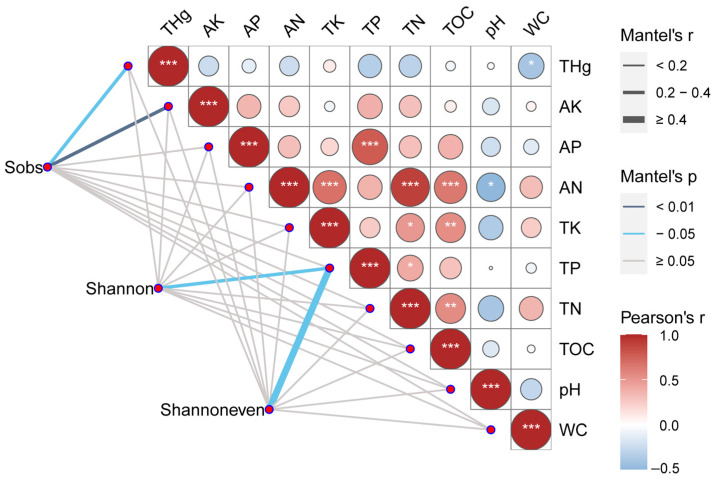
Correlation analysis between soil properties and the Sobs, Shannon, and Shannoneven indexes of arbuscular mycorrhizal fungi (AMF) community composition (A). Pairwise comparisons of environmental factors are shown with a color gradient denoting the Pearson correlation coefficients. The edge width corresponds to Mantel’s R statistic for the corresponding distance correlations, and the edge color denotes the statistical significance. *: *p* < 0.05; **: *p* < 0.01; ***: *p* < 0.001. THg: total mercury; AK: available potassium; AP: available phosphorus; AN: available nitrogen; TK: total potassium; TP: total phosphorus; TN: total nitrogen; TOC: total organic carbon; WC: water content.

**Figure 6 jof-09-00395-f006:**
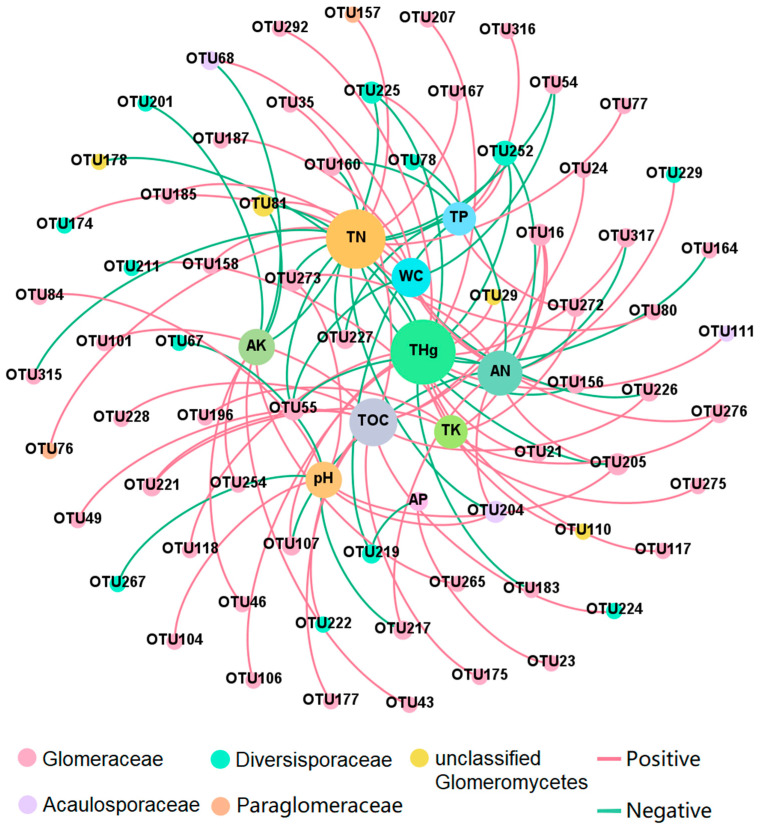
Two-factor network analysis of the association between arbuscular mycorrhizal fungi species and environmental factors of Hg mining area. The different colors of the species represent the family they belong to. The line between the points represents a significant correlation (*p* < 0.05) as indicated by the Pearson correlation coefficient. THg: total mercury; AK: available potassium; AP: available phosphorus; AN: available nitrogen; TK: total potassium; TP: total phosphorus; TN: total nitrogen; TOC: total organic carbon; WC: water content.

**Table 1 jof-09-00395-t001:** Significance levels (F values) of the effects of sampling sites and plant on Sobs, Shannon, Shannoneven, and Coverage indexes of arbuscular mycorrhizal fungi (AMF) based on a one-way analysis of variance (ANOVA).

Index	Group	SS	Df	MS	F Value	*p* Value
Sobs	Sampling sites	5597.500	6.000	932.917	6.742	0.001
	Plant	312.429	3.000	104.143	0.273	0.844
Shannon	Sampling sites	3.168	6.000	0.528	3.562	0.018
	Plant	0.250	3.000	0.083	0.306	0.821
Shannoneven	Sampling sites	0.092	6.000	0.015	2.513	0.063
	Plant	0.016	3.000	0.005	0.591	0.628
Coverage	Sampling sites	0.000	6.000	0.000	1.230	0.339
	Plant	0.000	3.000	0.000	1.601	0.221

Note: SS: sum of squares; Df: degree of freedom; MS: mean square.

**Table 2 jof-09-00395-t002:** Concentration range of soil variables, factor loadings, and explained variance following a principal component analysis (PCA) on the soil variables of samples from Hg mining area. The variables highly associated with particular PCs are indicated in bold.

Soil Properties	Content (Min–Max)	PC1	PC2	PC3
THg (mg/kg)	1.90–105.00	−0.334	−0.204	0.751
AK (g/kg)	0.169–0.607	0.400	0.459	−0.332
AP (mg/kg)	6.945–80.024	**0.579**	**0.633**	0.212
AN (mg/kg)	1.328–35.754	**0.918**	−0.257	0.038
TK (g/kg)	28.710–53.194	**0.663**	−0.409	0.364
TP (mg/kg)	438.597–415.663	**0.615**	**0.663**	0.017
TN (g/kg)	0.983–2.917	**0.874**	−0.161	−0.114
TOC (g/kg)	6.077–85.240	**0.683**	−0.058	0.414
pH	6.930–8.640	**−0.506**	0.363	−0.001
WC	0.148–0.266	0.316	−0.492	**−0.659**

Note: THg: total mercury; AK: available potassium; AP: available phosphorus; AN: available nitrogen; TK: total potassium; TP: total phosphorus; TN: total nitrogen; TOC: total organic carbon; WC: water content.

## Data Availability

Not applicable.

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
