# Peer review of "Soil Mercury Pollution Changes Soil Arbuscular Mycorrhizal Fungal Community Composition"

_jof, 2023, doi:10.3390/jof9040395_

Round 1
Reviewer 1 Report
Comments to the manuscript: Soil mercury pollution changes soil arbuscular mycorrhizal 2
fungal community composition
General comments:
This study aims to identify the influencing factors of soil AMF community composition in the Hg mining area. The authors should clearly state how they reach to understand how Hg affects AMF species richness and diversity, since they use DNA analysis instead of spore identities. Also, I think that it is necessary a hypothesis of this study.
Introduction
I feel that this section should include information about the use of dominant native plants species (v.g. Eleusine indica, Artemisia argyi, Astragalus sine, and Conyza canadensis) for phytoremediation, since authors used these species as a factor in this study.
There are no hypotheses in the introduction of this study, I recommend including it at the end of this section.
Methods
I don't see an analysis that would allow the authors to solve the third objective: understand how Hg affects AMF species richness and diversity. It would also have been important to analyze whether the roots of the dominant plants were colonized by AMF, since the identity of the plant was used as a factor influencing the richness and diversity of AMF.
Subtitle 2.3 Measurement of soil properties is repeated in lines 131 and 151, please correct.
Line 112: Please explain what does THg mean, since it is the first time it is reported.
Paragraph 152-175 does not have a subtitle, I assume it is Statistical analysis
Results
In this section, authors should avoid rewriting the methods used in the study, for example: lines 188-190 and 193-195.
Discussion
The authors must highlight how the aims of the work were approached. Authors should be more cautious when comparing their results with work in which fungal spores are used to calculate richness and diversity, since the use of DNA analysis (OTUs) is not equivalent.
Lines 353-354 this statement is speculative.
Author Response
Response to Reviewer 1 Comments
Dear reviewer,
We really appreciate your comments. We have accepted all the suggestions and revised the relevant parts. We hope this revision could improve the quality of this manuscript. The revised notes are as follows:
Point 1: This study aims to identify the influencing factors of soil AMF community composition in the Hg mining area. The authors should clearly state how they reach to understand how Hg affects AMF species richness and diversity, since they use DNA analysis instead of spore identities. Also, I think that it is necessary a hypothesis of this study.
Response 1: We appreciate the reviewer’s comment. We have made some modifications to illustrate how THg affects AMF richness and diversity (Lines 92–100, 270–270, 284–291 Figure 5, and 359–361). We then added the hypotheses for this study in the Introduction (Lines 90–92).
- “To address these hypotheses, this study (1) determined the dominant families, richness, and diversity of AMF in rhizosphere soils of dominant plant species ( indica, A. argyi, A. sinicus, and C. canadensis) in different sampling sites; (2) compared the differences in AMF richness, diversity, and community composition among different host plants and different sampling sites; (3) identified the principal components (PCs) of soil properties that significantly affected AMF community composition, and quantified the corresponding responses of AMF richness and diversity to elevated soil Hg content; and (4) recognized the correlations between the AMF of different taxa and different soil properties. ”
- “A Mantel test was applied to calculate the correlation between soil properties and the Sobs, Shannon, and Shannoneven indexes (Figure 5). TAs was significantly negatively correlated with WC, while TP had a significant positive correlation with AP. There were significant correlations among AN, TOC, TN, and TK, and they were negatively correlated with pH. Sobs index was correlated with THg (R = 0.227; p = 0.027) and AK (R = 0.314, p = 0.002); Shannon (R = 0.490, p = 0.012) and Shannoneven (R = 0.393, p = 0.030) indexes were correlated with TP. Further analysis of Pearson correlation coefficients between soil properties and AMF richness and diversity indexes revealed that THg had a significant negative correlation with the Sobs index and negatively but not significantly correlated with Shannon and Shannoneven indexes (Table S7). ”
- “Figure 5. Correlation analysis between soil properties and the Sobs, Shannon, and Shannoneven indexes of arbuscular mycorrhizal fungi (AMF) community composition (A). Pairwise comparisons of environmental factors are shown with a color gradient denoting the Pearson correlation coefficients. The edge width corresponds to Mantel's R statistic for the corresponding distance correlations, and the edge color denotes the statistical significance. *: p < 0.05; **: p < 0.01; ***: p < 0.001. THg: total mercury; AK: available potassium; AP: available phosphorus; AN: available nitrogen; TK: total potassium; TP: total phosphorus; TN: total nitrogen; TOC: total organic carbon; WC: water content”
- “The mantel test showed that THg was negatively correlated with the Sobs index and negatively correlated with Shannon and Shannoneven indexes (Fig. 5).”
- “This study hypothesized that (1) soil Hg contamination altered soil AMF community composition and negatively affected AMF diversity; and (2) AMF from different families have different sensitivity to Hg. ”
Point 2: Introduction
I feel that this section should include information about the use of dominant native plants species (v.g. Eleusine indica, Artemisia argyi, Astragalus sine, and Conyza canadensis) for phytoremediation, since authors used these species as a factor in this study.
There are no hypotheses in the introduction of this study, I recommend including it at the end of this section.
Response 2: We are very thankful for the reviewer’s comments. We have provided some information about the use of dominant native plant species and added some related references (Lines 81–87 and 585–593). The hypotheses for the study have also been added (Lines 90–92).
- “Four plants (Eleusine indica, Artemisia argyi, Astragalus sinicus, and Conyza canadensis) that are widely distributed in Wanshan Hg mining areas were selected as the target plants for this study. In addition, indica has been confirmed to be widely distributed in Hg-contaminated areas [49]. A. argyi and C. canadensis have higher tolerance of Hg, with a bioaccumulation factor for Hg up to 30% of A. argyi [50]. AMF can alleviate the As stress and regulate the growth condition of A. sinicus, but the resistance of A. sinicus to Hg is unknown[51]. ”
- “ Napaldet, J. T.; Buot, Jr., I. R. Absorption of lead and mercury in sominant aquatic macrophytes of Balili River and its implication to phytoremediation of water bodies. Trop. Life Sci. Res. 2020, 31, 19–32. http://doi.org/10.21315/tlsr2020.31.2.2
- Zhao, J. T.; Li, Y. Y.; Gao, Y. X.; Li, B.; Li, Y. F.; Zhao, Y. L.; Cai, Z. F. Study of mercury resistant wild plants growing in the mercury mine area of Wanshan district,Guizhou Province. Asian J. Ecotox. 2014, 9, 881–887. http://doi.org/10. 7524 /AJE. 1673-5897-20140515006 (in Chinese)
- Liu, Y.; Imtiaz, M.; Ditta, A.; Rizwan, M. S.; Ashraf, M.; Mehmood, S.; Aziz, O.; Mubeen, F.; Ali, M.; Elahi, N. N.; Ijaz, R.; Sha, L.; Cao, S.; Tu, S. Response of growth, antioxidant enzymes and root exudates production towards As stress in Pteris vittata and in Astragalus sinicus colonized by arbuscular mycorrhizal fungi. Environ. Sci. Pollut. Res. 2020, 27, 2340–2352. http://doi.org/10.1007/s11356-019-06785-5 ”
- “This study hypothesized that (1) soil Hg contamination altered soil AMF community composition and negatively affected AMF diversity; and (2) AMF from different families have different sensitivity to Hg. “
Point 3: Methods
I don't see an analysis that would allow the authors to solve the third objective: understand how Hg affects AMF species richness and diversity. It would also have been important to analyze whether the roots of the dominant plants were colonized by AMF, since the identity of the plant was used as a factor influencing the richness and diversity of AMF.
Response 3: We are very thankful for the reviewer’s comments. We performed a Mantel test of the correlations between soil properties and AMF richness and diversity indexes, and supported the trend of correlations between soil properties and AMF richness and diversity indexes by Pearson correlation coefficients (Lines 190–192, 270–280, and 284–291 Figure 5). We also carried out the identification of AMF infection structures, which were present in all plants. We have incorporated the relevant elements of this work into our research methodology (Lines 119–120).
- “The Mantel test was used to compare the correlations between soil properties and the Sobs, Shannon, and Shannoneven indexes of AMF community composition in R using the ‘ggcor’ package. ”
- “A Mantel test was applied to calculate the correlation between soil properties and the Sobs, Shannon, and Shannoneven indexes (Figure 5). TAs was significantly negatively correlated with WC, while TP had a significant positive correlation with AP. There were significant correlations among AN, TOC, TN, and TK, and they were negatively correlated with pH. Sobs index was correlated with THg (R = 0.227; p = 0.027) and AK (R = 0.314, p = 0.002); Shannon (R = 0.490, p = 0.012) and Shannoneven (R = 0.393, p = 0.030) indexes were correlated with TP. Further analysis of Pearson correlation coefficients between soil properties and AMF richness and diversity indexes revealed that THg had a significant negative correlation with the Sobs index and negatively but not significantly correlated with Shannon and Shannoneven indexes (Table S7). ”
- “Figure 5. Correlation analysis between soil properties and the Sobs, Shannon, and Shannoneven indexes of arbuscular mycorrhizal fungi (AMF) community composition (A). Pairwise compari-sons of environmental factors are shown with a color gradient denoting the Pearson correlation coefficients. The edge width corresponds to Mantel's R statistic for the corresponding distance correlations, and the edge color denotes the statistical significance. *: p < 0.05; **: p < 0.01; ***: p < 0.001. THg: total mercury; AK: available potassium; AP: available phosphorus; AN: available ni-trogen; TK: total potassium; TP: total phosphorus; TN: total nitrogen; TOC: total organic carbon; WC: water content. “
- “Arbuscular mycorrhizal colonization was detected in all the studied plant species. ”
Point 4: Subtitle 2.3 Measurement of soil properties is repeated in lines 131 and 151, please correct.
Response 4: We appreciate the reviewer’s comment and apologize for causing this problem. The error has been corrected (Lines 145, 165).
- “3 AMF community analysis ”
- “4 Statistical and data analysis ”
Point 5: Line 112: Please explain what does THg mean, since it is the first time it is reported.
Response 5: We appreciate the reviewer’s comment. The error has been corrected (Line 126).
- “Total Hg (THg) was measured using an Hg vapor meter (HGA-100, Beijing Haiguang Instrument Co., Ltd., Beijing, China).”
Point 6: Paragraph 152-175 does not have a subtitle, I assume it is Statistical analysis
Response 6: We appreciate the reviewer’s comment and apologize for causing this problem. The subtitle has been added (Line 165).
- “4 Statistical and data analysis”
Point 7: Results
In this section, authors should avoid rewriting the methods used in the study, for example: lines 188-190 and 193-195.
Response 7: We appreciate the reviewer’s comment and we have deleted them. After the modification, the relevant content has also been adjusted (Line 207–211, 214–216).
- “Almost all rarefaction curves tended to saturate at the chosen sequencing depth (Fig. S1). The AMF Sobs, Shannon, Shannoneven, and Coverage indexes were shown in Fig. S2.”
- “According to the result of one-way ANOVA (Table 1),”
Point 8: Discussion
The authors must highlight how the aims of the work were approached. Authors should be more cautious when comparing their results with work in which fungal spores are used to calculate richness and diversity, since the use of DNA analysis (OTUs) is not equivalent.
Response 8: We are very thankful for the reviewer’s comments. Through further reading of the paper, we have revised some ambiguous parts and deleted some contents. (Lines 369, 382–384).
- “High soil heavy metals concentrations can also inhibit spore germination and AMF mycelium growth, thereby reducing spore densities and AMF species abundances [41,60]”
Point 9: Lines 353-354 this statement is speculative.
Response 9: We appreciate the reviewer’s comment and found some evidence to support this speculative (Line 399).
“Finally, AMF diversity decreases due to the competition for carbohydrates between AMF species [63].”
Reviewer 2 Report
Line 101: Plot Units?
Describe briefly, in MyM, the Sobs, Shannon, Shannoneven, and Coverage indices. Any bibliographical references?.
Put the meaning of abbreviations to improve understanding of Tables and Figures.
Author Response
Response to Reviewer 2 Comments
Dear reviewer,
We really appreciate your comments. We have accepted all the suggestions and revised the relevant parts. We hope this revision could improve the quality of this manuscript. The revised note are as follows:
Point 1: Line 101: Plot Units?
Response 1: We appreciate the reviewer’s comment and apologize for causing this problem. We have added the unit of the plot (Line 114).
- “A 20 × 20 m2 plot with evenly distributed plant species and soil properties was selected at each sampling site.”
Point 2: Describe briefly, in MyM, the Sobs, Shannon, Shannoneven, and Coverage indices. Any bibliographical references?.
Response 2: We appreciate the reviewer’s comment. References to these indexes calculation methods have been added (Lines 168–171, 599–603).
- “Sobs is the observed values of richness. Shannon index was calculated according to Shannnon and Weaver [54]. Shannoneven index was calculated according to Pielou [55]. Coverage index was calculated according to Good [56].”
- “ Shannon, C.E., Weaver, W. The Mathematical Theory of Communication. University of Illinois Press: Urbana, Illinois, USA, 1949.
- Pielou, E.C. Ecological diversity. Wiley: New York, USA, 1975.
- Good, I. J. On the population frequencies of species and the estimation of population parameters. Biometrika. 1953, 40, 237–264. https://doi.org/10.2307/2333344”
Point 3: Put the meaning of abbreviations to improve understanding of Tables and Figures.
Response 3: We appreciate the reviewer’s comment and added the meaning of abbreviations of Tables and Figures (Lines 223, 253–255, 306–308, Supplementary material Lines 23, 30, 34–36, 51, 57–59).
